# Bioactivity of Wild and Cultivated Legumes: Phytochemical Content and Antioxidant Properties

**DOI:** 10.3390/antiox12040852

**Published:** 2023-04-01

**Authors:** Eleni D. Myrtsi, Epameinondas Evergetis, Sofia D. Koulocheri, Serkos A. Haroutounian

**Affiliations:** Laboratory of Nutritional Physiology and Feeding, Department of Animal Science, School of Animal Biosciences, Agricultural University of Athens, Iera Odos 75, 11855 Athens, Greece

**Keywords:** *Astragalus*, *Bituminoria bituminosa*, carotenoids, *Cicer*, *Fabaceae*, *Lathyrus*, phenolics, tannins, *Trifolium*

## Abstract

The global demand for increased meat production has brought to the surface several obstacles concerning environmental impacts, animals’ welfare, and quality features, revealing the need to produce safe foodstuffs with an environmentally acceptable procedure. In this regard, the incorporation of legumes into animal diets constitutes a sustainable way out that prevents these apprehensions. Legumes are plant crops belonging to the *Fabaceae* family and are known for their rich content of secondary metabolites., displaying significant antioxidant properties and a series of health and environmental benefits. The study herein aims to investigate the chemical composition and antioxidant activities of indigenous and cultivated legume plants used for food and feed. The respective results indicate that the methanolic extract of *Lathyrus laxiflorus* (Desf.) Kuntze displayed the highest phenolic (64.8 mg gallic acid equivalents/g extract) and tannin (419.6 mg catechin equivalents/g extract) content, while the dichloromethane extract of *Astragalus glycyphyllos* L., *Trifolium physodes* Steven ex M.Bieb. and *Bituminaria bituminosa* (L.) C.H.Stirt. plant samples exhibited the richest content in carotenoids lutein (0.0431 mg/g *A. glycyphyllos* extract and 0.0546 mg/g *B. bituminosa* extract), *α*-carotene (0.0431 mg/g *T. physodes* extract) and *β*-carotene (0.090 mg/g *T. physodes* extract and 0.3705 mg/g *B. bituminosa* extract) establishing their potential role as vitamin A precursor sources. Results presented herein verify the great potential of *Fabaceae* family plants for utilization as pasture plants and/or dietary ingredients, since their cultivation has a positive impact on the environment, and they were found to contain essential nutrients capable to improve health, welfare, and safety.

## 1. Introduction

The significant increase in the global population in combination with the improvement of economic standards for people living in overpopulated developing countries has created a global peak in the demand for meat and meat-containing foodstuffs [1,2] revealing the need for increasing the global volume of animal production. To meet this demand, the livestock husbandry sector farms were forced to expand [3] and transform their character from traditionally low-intensity nomadic rural farms to industrialized high-throughput estates. This transition has initiated a large circle of skepticism with respect to the environmental and climatic impacts derived from this global expansion of the animal production sector [4]. The above-mentioned reservations, combined with public concerns about the safety and quality features of the produced foodstuffs and animals’ welfare [5], have highlighted the nutritional health of livestock at the forefront of the European Union’s legislative framework [6]. This functional necessity is influenced by multiple parameters, such as food availability, feed nutritional quality, consumption, digestibility, and feed metabolism [7]. However, the observed recent disruption of global supply chains has put under question—or significant risk—these vital parameters for the achievement of sustainable production [8]. For this purpose, the utilization of legumes, both as forage and silage, in animal production systems provides a sustainable way out for the circumvention of this problem, since there are many literature reports indicating that legumes can perfectly be incorporated into livestock nutrition rations, exerting a positive effect on their productivity [9,10]. From a nutritional point of view, legumes are well known for their rich content of protein, fat, carbohydrates, and vitamins [11] that greatly contribute to livestock growth, along with various benefits for their healthy nutrients, in the form of natural phytochemicals. Among them, phenolic compounds constitute the most notable and studied class of molecules, mainly due to their pronounced antioxidant properties and numerous health benefits [12]. The majority of phenolic compounds found in legume beans are classified as phenolic acids, flavonoids, and condensed tannins [13]. It is also noticeable that the 20% of dry matter of forage legumes used in ruminant feeding consists of condensed tannins, which are responsible for many of their beneficial effects such as animal growth improvement, enhancement of milk production volume, and reduction of their methane emissions [14]. Carotenoids constitute an additional class of bioactive natural compounds found in legumes and are also known to display significant antioxidant activities along with pigment properties that establish their role as valuable nutritional additives associated with numerous health benefits [15,16].

Legumes belong to the Fabaceae family, formerly known as the family of Leguminosae, which comprise the third-largest family of angiosperms. Worldwide the Fabaceae family plants consist of more than 700 genera and almost 20,000 species [17], all abundant in the Mediterranean basin areas, subtropical savannahs, and tropical or dry forests [18,19]. Literature reports on their phytochemical content and antioxidant properties mostly concern their crops which consist of one of the main food and feed staples of humanity [15,20,21,22]. On the other hand, relative reports on silage-oriented legume crops are relatively scarce considering the number of indigenous legumes found in natural pastures and meadows [23]. These studies are mainly focusing on the presence and distribution of various phytochemicals in legume foods and feed seeds, with the most celebrated research subject as the assessment of phenolic compounds content, accounting for 3808 studies for more than 125 legume *taxa* [24,25,26,27,28,29,30]. The determination of their tannin and carotenoid content has attracted significantly less attention with, respectively only 2000 and 633 studies [31,32,33]. A similar motif is also observed for studies concerning the exploitation of their antioxidant properties, which are limited to the evaluation of 50 Fabaceae plant *taxa* [25,32,33,34]. 

The main objective of the research presented herein is the implementation of a comparative investigation of indigenous and cultivated legume plants used for food and feed, with respect to the phytochemical content and antioxidant properties of their vegetative parts. This endeavor aspires to partially cover the identified knowledge gap between the widely studied legume crops and their indigenous wild relatives, aiming to identify either some promising new crops or the potential benefits derived by the wild foraging of livestock in the Mediterranean pastures.

## 2. Materials and Methods

### 2.1. Materials

#### 2.1.1. Plant Material and Extracts

Plant material was retrieved from eight different species of the Fabaceae family. All samples consisted of whole plants, which were collected in their vegetative stage of late flowering-early seed development. The *taxa* collected are included in Table 1, along with their localities, year of collection, and extraction yields. A voucher specimen of each plant collected has been deposited in the herbarium of the Agricultural University of Athens, Greece. 

#### 2.1.2. Chemicals and Standards

Solvents used for plant samples extraction: dichloromethane (DCM, analytical purity; Fisher Chemicals, Hampton, New Hampshire (NH), USA), hexane (Hex, analytical purity; Fisher Chemicals, Hampton, NH, USA), and methanol (MeOH, analytical purity; Fisher Chemicals, Hampton, NH, USA).

Carotenoid standards used for the determination of their qualitative and quantitative presence: astaxanthin (≥95% purity; Sigma-Aldrich, St. Louis, Missouri (MO), USA), fucoxanthin (≥95% purity; Sigma-Aldrich, St. Louis, MO, USA), lutein (≥95% purity; ExtraSynthese, France, Lyon), lycopene (≥95% purity; Sigma-Aldrich, USA, St. Louis, Missouri), phytoene (≥95% purity; CaroteNature, Münsingen, Switzerland), α-carotene (≥95% purity; CaroteNature, Münsingen, Switzerland), β-carotene (≥95% purity; Sigma-Aldrich, St. Louis, MO, USA), and β-cryptoxanthin (≥95% purity; ExtraSynthese, Lyon, France). All standards used for the determination of phenolic compounds were provided by Sigma-Aldrich (≥95% purity; St. Louis, MO, USA), except from catechin, epigallocatechin gallate, gallic acid isoquercetin (≥95% purity; ExtraSynthese, Lyon, France).

Solvents used for the LC-MS/MS (Liquid Chromatography-Mass Spectrometry) determinations: acetonitrile (LC-MS purity; JT Baker, Phillipsburg, New Jersey (NJ), USA), formic acid (LC-MS purity; Fisher Chemical, Hampton, NH, USA), methanol (LC-MS purity; JT Baker, Phillipsburg, NJ, USA) and water (LC-MS purity; JT Baker, Phillipsburg, NJ, USA).

Reagents and solvents used for assessments: 2,2-diphenyl-1-picrylhydrazyl (DPPH, (≥95% purity; Sigma-Aldrich, St. Louis, MO, USA), 2,4,6-tris(2-pyridyl)-s-triazine (TPTZ, ≥95% purity; Sigma-Aldrich, St. Louis, MO, USA), anhydrous sodium carbonate (99.999+% purity; Chem-Lab, Zedelgem, Belgium), chloroform (analytical purity; Fisher Chemicals, Hampton, NH, USA), diethyl ether (analytical purity; Fisher Chemicals, Hampton, NH, USA) dimethylsulfoxide (DMSO, analytical purity; Fisher Chemicals, Hampton, NH, USA), Folin–Ciocalteu’s reagent (2 N; Sigma-Aldrich, St. Louis, MO, USA), glacial acetic acid (analytical purity; Sigma-Aldrich, St. Louis, MO, USA), heptahydrate iron sulfate (99+% purity; Alfa Aesar, Stoughton, Massachousetts (MA), USA), hexahydrate ferric chloride (97% purity; Alfa Aesar, Stoughton, MA, USA), petroleum ether (40–60 °C) (analytical purity; Fisher Chemical, Hampton, NH, USA), Potassium hydroxide (KOH, 99.99% purity; Alfa Aesar, Stoughton, MA, USA), sulfuric acid (98% purity; Chem-Lab, Zedelgem, Belgium), Trolox (6-hydroxy-2,5,7,8-tetramethylchroman-2-carboxylic acid (95% purity; Acros Organics, Geel, Belgium), vanillin (99% purity; Acros Organics, Geel, Belgium).

#### 2.1.3. Equipment

Extract condensation was performed under vacuum and heat-assisted evaporation, in temperatures below 35 °C, using a Büchi Rotavapor R-210 apparatus, equipped with a Büchi vacuum pump V-700, Vacuum controller V-850 (all obtained from Büchi, Flawil, St Gallen, Switzerland), and Julabo F12 (Seelbach, Germany) cooling unit. 

The estimation of Total Phenolic Content (TPC) and Total Tannin Content (TTC) was implemented using an Infinite^®^ 200 PRO microplate reader (Tecan Group Ltd., San Jose, CA, USA) and the estimation of Total Carotenoid Content (TCC) was performed using an x-ma 100 spectrophotometer (Human Corporation, Seoul, Republic of Korea).

An Accela Ultra High-Performance Liquid Chromatography system equipped with an autosampler and coupled with a TSQ Quantum Access triple-quadrupole mass spectrometer (Thermo Fisher Scientific, Inc., Waltham, MA, USA) was used for the determination of the phenolic and carotenoid compounds qualitative and quantitative content.

### 2.2. Methods

#### 2.2.1. Extraction Procedure

After collection, each plant material was allowed to dry in a shady, well-ventilated covered area. Then, they were ground and kept in special containers until extraction.

All samples were extracted successively with n-hexane, dichloromethane, and methanol. Each extraction lasted 48 h and was repeated in triplicate for each solvent using seven parts of solvent for one part of the sample. The extracts obtained were condensed under vacuum and heat-assisted evaporation in temperatures below 35 °C. The respective yields of the extractions are presented in Table 1.

#### 2.2.2. Phytochemical Analysis 

All measurements were performed in triplicate and the respective results were recorded as the mean ± standard deviation of three replicates.

##### Total Phenolic Content (TPC)

TPCs of all samples were measured using a modified version of a previously reported spectrophotometric method [35]. Specifically, 10 μL of each methanolic extract was diluted in 100 μL of water and Folin–Ciocalteu reagent solution (10 μL) was added to a 96-well microplate in triplicate (Sarstedt AG & Co. KG, Nümbrecht, Germany) and incubated for 3 min at room temperature. Then, 20 µL of sodium carbonate aqueous solution (7.5% *w*/*v*) and 60 μL of water were added and the incubation was continued in the dark for an additional 60 min. Each sample’s absorption was measured at 765 nm wavelength and the value was used for the determination of its TPC using a standard calibration curve previously prepared with 30–180 µg/mL solutions of gallic acid in methanol. The results were obtained by applying the equation y = 0.0038258x − 0.0090575 (R^2^ = 0.99996) and are expressed as mg of gallic acid equivalents per g of extract (mg GAE/g of extract).

##### Total Tannin Content (TTC)

TTC values of methanolic extracts were determined in accordance with a modified version of a literature spectrophotometric method [35]. Briefly, 25 µL of each extract were mixed with 150 µL of a methanolic vanillin solution (4% *w*/*v*), and an equal volume of a methanolic solution of sulfuric acid (32%) was added. The mixture was placed into a 96-well microplate in triplicate and incubated at room temperature in the dark for 15 min. The absorbance of each solution was measured at 500 nm wavelength and the TTC of the sample was determined against a standard calibration curve of catechin solutions at concentrations ranging from 150 to 1200 µg/mL. The respective results were obtained using the calibration curve *y* = 7.261e – 005*x* + 0.048697 (R^2^ = 0.99985) and are expressed as mg of catechin equivalents per g of extract (mg CE/g extract). 

##### Total Carotenoid Content (TCC)

TCCs of hexane and dichloromethane extracts were determined following a modified version of a previously described spectrophotometric method [36]. In specific, 1 mg of the hexane and dichloromethane extract of each plant sample was diluted in 1 mL hexane and dichloromethane solvent, respectively, and the absorbance was measured at 450 nm wavelength. The TCC was determined using a standard calibration curve of *β*-carotene solutions at concentrations ranging from 0.25 to 4.0 µg/mL. All measurements were performed in triplicate and results were calculated using the calibration curve equation *y* = 0.2105*x* – 0.0022 (R² = 0.9996). Respective results are expressed as mg of *β*-carotene equivalents per g of extract (mg *β*-CE/g extract). 

##### Phenolic Compounds Fingerprinting

The detailed determination of the individual phenolic compound’s presence was performed with an LC-MS/MS system, operated using an analytical method previously described by Myrtsi et al. [35]. The following compounds were used as standards: gallic acid, protocatechuic acid, epigallocatechin, procyanidin B1, chlorogenic acid, procyanidin B2, catechin, epicatechin, epigallocatechin gallate, caffeic acid, herperidin, polydatin, isoquercetin, rutin, *p*-coumaric acid, sinapic acid, myricetin, coniferyl aldehyde, resveratrol, quercetin, apigenin, kaempferol, and isorhamnetin. Their separation was achieved on a 150 × 2.1, 3 μm reverse phase column obtained from Fortis Technologies Ltd. (Νeston, UK), connected with an AF C18 guard column (10 × 2.0 mm, 3 μm) (Fortis Technologies Ltd., Νeston, UK). The mobile phase consisted of water/formic acid (0.1%) mixture (Solvent A) and acetonitrile/formic acid (0.1%) (Solvent B). For the MS/MS determination, the ElectroSpray Ionization (ESI) technique was utilized in Selected Reaction Monitoring (SRM) mode. The analytes detected were determined using the following equations: procyanidin B1: *y* = −0.00258365 + 0.0256469*x* (R^2^ = 0.9998), chlorogenic acid: *y* = −0.00016251 + 0.0821949*x* (R^2^ = 0.9992), procyanidin B2: *y* = −0.00111717 + 0.0215283*x* (R^2^ = 0.9996), epicatechin: *y* = 0.0000486191 + 0.00498918*x* (R^2^ = 0.9989), epigallocatechin gallate: *y* = 0.000118646 + 0.00994617*x* (R^2^ = 0.9993), hesperidin: *y* = 0.000321812 + 0.000883822*x* (R^2^ = 0.9988), isoquercetin: *y* = −0.000728175 + 0.0484013*x* (R^2^ = 0.9998), rutin: *y* = −0.00209729 + 0.0613716*x* (R^2^ = 0.9997), quercetin: *y* = −0.00140527 + 0.068703*x* (R^2^ = 0.9999), apigenin: *y* = 0.00884981 + 0.300563*x* (R^2^ = 0.9993), kaempferol: *y* = −0.000290336 + 0.0430157*x* (R^2^ = 0.9995). The respective results are expressed as mg of the phenolic compound per g of extract. 

##### Carotenoids Fingerprinting

The carotenoid quantitation was accomplished using the same LC-MS/MS system connected with an APCI (Atmospheric Pressure Chemical Ionization) source in SRM mode. The carotenoids separation and detection were performed in accordance with an analytical method developed by Myrtsi et al. [37], using the following compounds as standards: fucoxanthin, astaxanthin, lutein, *β*-cryptoxanthin, lycopene, *β*-carotene, α-carotene, and phytoene. The separation was achieved on a 120 EC-C18 reverse phase column (2.1 × 100 mm, 1.9 µm), obtained from Infinity Lab Poroshell-Agilent, Santa Clara, CA, USA, with a 120 EC-C18 guard column (2.1 × 5 mm, 2.7µm) (Infinity Lab Poroshell-Agilent, Santa Clara, CA, USA). The mobile phase consisted of acetonitrile (Solvent A), water (Solvent B), and MeOH (Solvent C). The detected analytes were determined with the equations: lutein: *y* = 89,199.8 + 1.61762*e* + 06*x* (R^2^ = 0.9988), *β*-cryptoxanthin: *y* = −1104.48 + 42,508.5*x* (R^2^ = 0.9994), α-carotene: *y* = −141,767 + 1.23176*e* + 06*x*, (R^2^ = 0.9988), *β*-carotene: *y* = 61,639.6 + 348546*x* (R^2^ = 0.9995). The results are expressed as mg of each carotenoid per g of extract.

#### 2.2.3. Antioxidant Properties Evaluation

The antioxidant capacities estimation of the samples studied was performed using the Ferric Reducing Antioxidant Power (FRAP) and the 2,2-diphenyl-1-picrylhydrazyl (DPPH) assays. All experiments were performed in triplicate and the respective results were presented as the mean ± standard deviation of the three replicates.

##### Ferric Reducing Antioxidant Power (FRAP) Assay

The reducing capacity of samples was determined in accordance with a literature method [35] that records the ability of each sample to reduce the Fe^3+^ of a Fe^3+^-TPTZ complex (ferric-2,4,6-tripyridyl-s-triazine) into Fe^2+^-TPTZ. Briefly, just before the analysis, a FRAP reagent was prepared by mixing 10 mL of acetate buffer (pH 3.6) with 1 mL of 2,4,6-tris(2-pyridyl)-s-triazine (TPTZ) (10 mM in HCl 0.04 N) and 1 mL of ferric chloride hexahydrate (20 mM in distilled water). The mixture was placed in a water bath and the temperature was adjusted to 37 °C. Then, 30 μL of each sample methanolic extract was pipetted into a 96-well microplate, and 180 μL of FRAP reagent was added. The microplate was incubated in the dark for 30 min at 37 °C and the resulting solution absorbance was measured at 593 nm wavelength. The reducing capacity was determined against a standard calibration curve constructed (*y* = 1.9995*x* + 0.12137, R^2^ = 0.9994) for FeSO_4_ concentrations ranging from 0.005 to 0.100 mol/L. The respective results are expressed as mol Fe^2+^/kg of dry material.

##### DPPH • Radical Scavenging Assay

The DPPH• assay used for the evaluation of the radical scavenging activity has been described previously [35]. Briefly, 30 μL of each methanolic extract was placed into a 96-well microplate and 175 µL of a methanolic solution of DPPH• radicals (0.1 M) were added. The mixture was incubated at room temperature for 40 min. Then, the absorbance of the mixture was measured at 515 nm wavelength and the antioxidant activity was determined against a calibration standard curve, constructed with methanolic solutions of Trolox at concentrations ranging from 100 to 280 µg/mL. The respective results are calculated by the calibration curve equation *y* = 0.3485*x* + 5.7089 (R^2^ = 0.9992) and they are expressed as mg Trolox Equivalents (TE)/kg of dry material.

### 2.3. Statistical Analysis

All results are presented as mean value ± standard deviation (SD) of experiments performed in triplicate. For all calculations performed in this work, the Durbin–Watson (DW) statistical tests for the residuals and the οne-way analysis of variance (ANOVA) were used and the table indicated that the *p*-value was always less than 0.05. Τhe statistical functions of Microsoft Office 365 ANOVA (Microsoft, Redmond Campus, Washington DC, USA) were used. Bonferroni tests for the *p*-values correlation were performed using the Stata17 program (StataCrop LLC, College Station, TX, USA). The *p*-values correlation between samples was performed with Bonferroni tests and the respective results are included as Appendix A.

## 3. Results

### 3.1. Phytochemical Analysis

The applied herein fractional extraction protocol furnished extracts with differentiated phytochemical profiles because the extraction with methanol yielded the separation of contained polyphenols and tannins, while the extractions with hexane and dichloromethane-solvents provided carotenoid-rich extracts.

#### 3.1.1. Phenolics

The outcome of the assessment of the presence and quantitation of individual phenolics into methanolic extracts is included in Table 2, along with their TPC and TTC values, while their *p*-values are provided as Appendix A. Results herein constitute the first detailed insight into phenolic content for these species and include the fingerprinting of 22 phenolic compounds. In total, the presence of 11 phenolics was revealed in various plant samples, while the remaining compounds (isorhamnetin, *p*-coumaric acid, sinapic acid, myricetin, resveratrol, polydatin, caffeic acid, catechin, gallic acid, protocatechuic acid, epigallocatechin) were not detected in samples studied.

#### 3.1.2. Carotenoids

The results of the assessment of carotenoids’ presence in hexane and dichloromethane extracts, along with their TCCs determinations, are presented in Table 3 and constitute the first report and detailed fingerprinting of eight major carotenoids present in these plants. Their *p*-values are provided as Appendix A. In particular, the presence of four carotenoids was revealed while the remaining four, namely fucoxanthin, astaxanthin, lycopene, and phytoene were not detected in these samples.

### 3.2. Antioxidant Properties Determination Assays

The antioxidant activity of the methanolic extracts was evaluated through the performance of DPPH and FRAP assays. The respective results are depicted in Table 4, representing the first-time report of the antioxidant properties of these plants. The respective *p*-values are provided as Appendix A.

## 4. Discussion

### 4.1. Phytochemical Analysis

#### 4.1.1. Phenolic Compounds Content

In general, among all methanolic extracts exploited those of *L. laxiflorus* were determined to display the richest phytochemical content and exhibited the highest TPC and TTC values. Although both studied *Trifolium* methanolic extracts were determined to display comparable TPC values, the extract originated from wild *T. physodes* was totally deprived of tannins, which were detected in moderate concentration in the cultivated specimen of *T. repens*. A similar pattern was also observed for the two *Astragalus* extracts, since the extract of *A. creticus* displayed a particularly high tannin content, while tannins were not detected in the extract of *A. glycyphyllos*. It is also noticeable that both lacking tannins specimens (*T. physodes* and *A. glycyphyllos*) share the same natural habitat and grow under the canopy of *Abies cephallonica*. Their common origin, in relation with the absence of tannins, is worthwhile of further investigation. On the other hand, the respective results for two *Cicer* extracts provided a further clause to the previous hypothesis, since they displayed comparable TPC and TTC values although *C. incisum* constitutes a wild species and *C. arientinum* a cultivated species, decoupling this differentiation from their tannin content. Finally, *Bituminaria bituminosa* extract proved as a rather unfavorable source for both tannins and phenolic compounds.

A thorough literature review of previous research results concerning the studied *Fabaceae* plants verified that the TPC and TTC values for *T. physodes* and *C. incisum* extracts are reported herein for first-time. A similar observation also applies to the determination of *L. laxiflorus* TPC [38], *A. creticus* TPC [39], and TTC [40], *C. arientinum* TTC [41], *B. bituminosa* [42,43], and *A. glycyphyllos* TPC [44]. It must be noted that *T. repens*, a widely cultivated forage crop, comprises the only well-documented plant in the literature with numerous studies. In addition, studies concerning the presence of individual polyphenols in legume plant tissue are also limited, mostly referring to their isoflavone content [44,45,46]. Their majority is focused on *Trifolium* species and especially *T. pratense* and *T. repens* specimens [47]. The study herein includes a thorough polyphenolic fingerprinting of all plants studied, all presented for first time in the literature. It must be noted, however, that a comparative discussion of TPC and TTC values reported is challenging since the assessments of these variables available in previous studies have been implemented using a broad variety of enumeration scales.

Nevertheless, the research of Spanou et al. [38] on the methanolic extract of *L. laxiflorus* aerial parts has determined TPC value of 89 mg GAE/g, which is comparable with our finding of 64.8 mg GAE/g methanolic extract. On the contrary, the reported herein TTC value of 419.6 mg CE/g for the same extract differs significantly from those reported for condensate tannin content of *L. sativus* seeds, which vary from 890 to 5180 mg/kg of dry matter [48]. It must be noted however, that considering the extraction yields of Table 1, the reported herein TTC value corresponds to 41,630 mg CE/kg of dry matter, indicating that the tannin content of *Lathyrus* sp. herb is 8-fold higher as compared to tannins of the respective seeds.

The *Trifolium* sp. baseline of TPC content has been drawn by the respective results on the widely cultivated *T. pratense* and *T. repens* specimens [49,50]. In particular, the TPC value for *T. repens* flowers extract is calculated as 192.0 mg GAE/g extract [50], a value significantly higher than the reported herein TPC values of 24.228 and 27.7 mg GAE/g for *T. repens* and *T. physodes*, pointing out their flowers as the primary source of phenolics. This assumption is reinforced by reports for the TPC of *T. pratense* ranging from 44.4 to 53.7 mg/kg of dry matter [40], values in line with present findings adjusted in accordance with their extract yield. The amounts of condensed tannins detected in seeds, flowers, and leaves of *T. repens* highlighted flowers having the highest content (13–79 mg/g) and leaves with poorest (not detected–0.6 mg/g). Our results reflecting the extract of the whole plant, and corrected with the extraction yields, register TTC value falling within those determined for the two parts [51].

Another extensively studied *Fabaceae* plant is *Cicer arientinum*, widely cultivated as both food and fodder [52,53]. Its beans are known as the primary source of polyphenols and exert TPC value of 43,650 mg GAE/kg of legume flour [53]. In our study, we have measured, for the first time, the TPC value of the herb as 125.2 mg GAE/kg of dry matter (after inclusion of the extraction yield). On the other hand, previously reported TTC values for *C. arientinum* ranging from 2 to 3 mg CE per g of dry matter [41] are significantly lower as compared to values reported herein, presumably due to the selection of varieties with low tannin content.

*Bituminaria bituminosa*, a plant widely distributed in the Mediterranean basin, is well known to display a rich content of secondary metabolites [42]. Its literature TPC and TTC values have been calculated indirectly using a modelling approach, ranged, respectively from 7900–13,600 and 700–5000 mg/kg of dry matter [43]. These figures differ greatly from values determined herein as 1826 and 85 mg/kg of dry matter for TPC and TTC, respectively, raising serious doubts concerning the accuracy of calculated with the modelling approach.

Finally, for the rarely studied plants of the *Astrgalus* genus, there are only literature reports on the phenolic content of *A. creticus* [39] and *A. glycyphyllos* [46] and one, not quantified, report [40] for the tannin content of *A. creticus*. The respective TPC values are 79.82 mg GAE/g dry material for *A. creticus* methanolic extract, while for the ethanolic-aqueous extracts of *A. glycyphyllos* at flowering and branching stages are, respectively 18,500 and 17,100 mg GAE/kg of dry matter [44]. These figures are subtracted of value measured herein for the *A. glycyphyllos* methanolic extract as 23,784.3 mg/kg of dry matter. On the contrary, the literature value for *A. creticus* extract greatly excels of value determined in our study.

In terms of the importance of polyphenols and tannins content into *Fabaceae* plants, it must be noted that the tannin content of pasture plants greatly affects their grazing preferability since condensed tannin content exceeding 5,000 mg/kg of dry matter is found to deter animals [43]. In this respect, the extracts of the following four pasture plants *A. creticus*, *L. laxiflorus*, *C. incisum* and *T. repens* have been determined to contain increased tannins, presumably as plant response to grazing pressures. In addition, an increased tannin diet of beef steers is determined to reduce considerably their excretions of greenhouse gases [54].

Similar beneficial effects are also attributed to the presence of various phenolics such as the prevailing in most samples studied molecules of rutin and quercetin. Quercetin, the major constituent of *T. physodes* and *C. incisum* plant extracts (respectively 0.86 and 2.3 mg/g methanolic extract), has been extensively included as a supplement in broiler chickens’ basal diet because of its well-known ability to improve their growth performance traits and significantly increase the European Production Efficiency Factor (EPEF) [55]. Additionally, the beneficiary effects of long-term dietary supplementation of dairy cows with rutin are already established for Chinese Holstein cows, revealing its ability to increase their milk yield [56]. Thus, the presence of large amounts of rutin in *A. creticus* and *L. laxiflorus* extracts studied (respectively 1.8 and 7.79 mg/g methanolic extract) is indicative of their potential to be included in animal feeding diets. Similar results were also obtained for *L. laxiforus* and *T. repens*, which provided extracts containing the highest concentrations of natural polyphenols. Finally, it is noticeable that both plants contain chlorogenic acid, which has been determined to promote animal health and enhance the quality of meat products [57].

#### 4.1.2. Carotenoid Compounds Content

The Total Carotenoid Contents (TCC) were estimated for both hexane and dichloromethane extract of plant samples, while the fingerprinting of carotenoid content was performed only for the DCM extracts which exhibited the highest TCC values. The plants displaying the highest TCC values are *A. glycyphyllos* (TCC_HEX_ + TCC_DCM_ = 23.76 mg β-CE/g extracts), *T. physodes* (TCC_HEX_ + TCC_DCM_ = 21.96 mg β-CE/g extracts) and *B. bituminosa* (TCC_HEX_ + TCC_DCM_ = 19.46 mg β-CE/g extracts). On the other hand, the LC-MS/MS analyses results of plant extracts revealed the presence of four carotenoids, namely α- and β-carotene, β-cryptoxanthin, and lutein. The largest concentrations of α-carotene were found in comparable quantities in *A. creticus*, *T. physodes*, and *T. repens* extracts, while the richest content of β-carotene displayed the *B. bituminosa* extract, which also exerted the highest lutein content. Finally, β-cryptoxanthin was detected in the extracts of *T. physodes* and both *L. laxiflorus* and *T. repens*.

Among the eight taxa studied, *A. glycyphyllos*, *A. creticus*, *T. physodes*, *L. laxiflorus*, and *C. incisum* are exploited herein for the first time herein for their carotenoid content. Among them, *A. creticus* and *T. physodes* were determined to display the richest carotenoid content, composed of lutein, α-carotene, and β-carotene.

The literature knowledge for the carotenoid content of *T. repens* has been provided by Elgersma et al. [58] who highlighted the plant as a good source of both β-carotene and lutein. Present results verified the presence of β-carotene, α-carotene, and β-cryptoxanthin in the same plant extract, with most abundant the molecule of β-carotene. With respect to the chickpea seeds, which are known to contain β-carotene and β-cryptoxanthin [30], in specific Razaei et al. [59] study on different variations of *C. arientinum* beans indicated the presence of lutein (10.6–45.9 mg/kg), zeaxanthin (8.1–31.8 mg/kg), β-carotene (1.7–2.6 mg/kg), β-cryptoxanthin (0.8–2.9 mg/kg), and violaxanthin (0.7–2.5 mg/kg). The herein detected presence of β-carotene in *C. arientinum* plants at a lower concentration as compared to the respective value reported for seeds. This differentiation can be rationalized by considering the different matrixes used for these studies. It must also be noted that within the *C. insicum* plant extract the studied carotenoids were not detected. Finally, Chaumont and Gudin have reported [60] TTC values for *B. bituminosa* leaves and calluses 2390 and 300 mg/kg, respectively. Our finding is closer to the callus value, while our LC-MS/MS analysis indicates that *B. bituminosa* plant extract comprises the richest source of lutein and β-carotene among the extracts studied.

The importance of carotenoids presence is indicated by the characterization of these compounds as the main precursor of vitamin A. The latter constitutes an essential component for all vertebrate animals’ diets, whose needs are mostly met through feeding with rations rich in provitamin A carotenoids, such as β-carotene [61]. Vitamin A is also essential for the ruminants’ diet [62]. Therefore, animal feeding with pasture plants containing carotenoids improves their health, welfare, and safety. Carotenoid-rich animal feed is also associated by Xu et al. [63] with an increased yield of milk with high lactose and fat content and the enhancement of antioxidant capacity. Results presented herein pointed out the plants of *A. glycyphyllos*, *T. physodes* and *B. bituminosa* as the best source of carotenoids, emerging their potential role as pasture plants in animal feeding.

### 4.2. Antioxidant Assays

The DPPH and FRAP assays were used for the estimation of the antioxidant capacities of all extracts studied. It is noticeable that in both assays the highest antioxidant activity was affirmed for the dichloromethane extracts.

The antioxidant capacities of the ethanolic extracts of various *Fabaceae* plants have been measured using the DPPH assay by Butkuté et al. [44]. In specific, they studied the antioxidant activities of two *Trifolium* sp. (*T. pratense* and *T. medium*) and two *Astragalus* sp. (*A. glycyphyllos* and *A. cicer*) and determined that *Trifolium* species display higher antioxidant activity. Herein, the DPPH assessment of the antioxidant capabilities of *T. physodes*, *T. repen*, *A. glycyphyllos*, and *A. creticus* methanolic extracts also proved the efficiency of *Trifolium* extracts. The antioxidant potential of *Trifolium* sp. was also studied by Jakubczyk et al. for the ethanolic extracts of *T. repens* and *T. pratense* flowers using both DPPH and FRAP assays [50]. The respective results for *T. repens*, expressed as a percentage of DPPH radical inhibition, was 3.86 ± 0.16%, while the reductive potential measured by FRAP was determined as 88.71 ± 3.16 µM Fe(II)/L.

Recently, the antioxidant properties of B. bituminosa leaves were studied through the DPPH assay for both hexane and methanolic extracts, highlighting hexane extract as the most efficient [64]. The examined herein plant extracts exhibited DPPH values of 16.1 and 5.6 mg Trolox Equivelant (TE)/g for hexane and methanolic extracts, respectively, while the dichloromethane extract was determined as the most potent exhibiting DPPH value of 21.04 mg TE/g extract.

Most previous studies on the antioxidant properties of *C. arientinum* mostly concern their seeds, which are a traditional Mediterranean food. Among them the most recent, in 2022, the study of Sharma and Giri [53] reported IC_50_ (half maximal Inhibitory Concentration) value for the DPPH assay of chickpeas methanolic extract 59.88 ± 0.26 μg/mL, using ascorbic acid as standard. Herein the methanolic extract of the whole plant displayed a particularly low DPPH value indicating that *C. arientinum* seeds constitute the plant part with the most potent antioxidant capability.

In regard to the results of the DPPH assay, the most effective sample was the methanolic extract of *Lathyrus laxiflorus*. This extract also displayed the highest content of chlorogenic acid (1.13 mg/g), procyanidin B2 (2.23 mg/g), epicatechin (1.52 mg/g), epigallocatechin gallate (1.883 mg/g), and rutin (7.79 mg/g), compounds which are repetitively reported as very efficient antioxidants [65,66,67,68].

The inclusion of feed additives rich in the form of antioxidant compounds in animal diets has not been widely studied, and is mostly limited to vinification by-products and soybean into broiler chickens’ diets in order to supply broilers with bioactive metabolites that are proven capable to improve their oxidative status and the oxidative stability of their products [69]. Thus, various *Fabaceae* plants and especially *T. repens* and *L. laxiflorus* which were determined herein to contain larger amounts of antioxidants can be included in animals’ diets to improve their health and productivity. Additionally, a wider utilization of *Fabaceae* plants can make a significant and multidimensional contribution to arable rotations with respect to the future requirement to produce large quantities of foods as consequences of the rapidly growing population. McKenna et al. [70] have already demonstrated the utility of *Trifolium pratense* plantation for soil fertility-building and highlighted the multiple benefits provided by its cultivation, such as the atmospheric nitrogen fixation, soil conservation, structural soil improvement and several agroecosystem services including the increase of soil microbial activity, the provision of food for pollinators, the phytoremediation of polluted soils, and the allelopathic activity. These benefits can be expanded to most of the studied *Fabaceae* plants, increasing the respective economic outcome because of their rotation profitability in terms of silage value and yield benefits.

## 5. Conclusions

Eight plants belonging to Fabaceae family were thoroughly investigated with respect to their antioxidant capacities and the assessment of bioactive natural phenolic, tannin, and carotenoid phytochemicals content. Most of these determinations are reported for the first time in the literature, aiming to highlight the potential of these plants for utilization as dietary ingredients. Most samples studied displayed a rich content of phytochemicals and potent antioxidant properties, revealing the usefulness of these plants for inclusion in livestock and human diets. Among them, Lathyrus laxiflorus (Desf.) Kuntze displayed the richest phytochemical content and exhibited the highest TPC and TTC values. This plant also exhibited the best antioxidant activity, presumably because of its rich content of chlorogenic acid, procyanidin B2, epicatechin, epigallocatechin gallate, and rutin, compounds known to exert significant antioxidant properties. On the other hand, dichloromethane extracts of Astragalus glycyphyllos L., Trifolium physodes Steven ex M.Bieb. and Bituminaria bituminosa (L.) C.H.Stirt. plant samples exhibited the richest content in carotenoids lutein, β-cryptoxanthin, α-carotene, and β-carotene establishing their potential role as vitamin A precursor sources. Results presented herein verify the great potential of these plants for utilization as pasture plants and/or dietary ingredients, since their cultivation has a positive impact on the environment, and they were found to contain essential nutrients capable to improve health, welfare, and safety.

## Figures and Tables

**Table 1 antioxidants-12-00852-t001:** Samples’ composition and extraction yields.

Sample	*Taxa*	Location	Yield (%)		
			Hexane	DCM	MeOH
L01 ^1^	*Astragalus creticus* Lam.	Parnassos Mt., 2017	0.53	0.54	12.3
L02 ^1^	*Astragalus glycyphyllos* L.	Parnassos Mt., 2017	1.81	1.57	12.6
L03 ^1^	*Lathyrus laxiflorus* (Desf.) Kuntze	Parnassos Mt., 2017	5.31	0.74	9.9
L04 ^1^	*Trifolium physodes* Steven ex M.Bieb.	Parnassos Mt., 2017	0.2	0.8	11.9
L05 ^1^	*Cicer incisum* (Willd.) K.Maly	Crete Isl., 2018	0.26	1.12	4.6
L06 ^1^	*Bituminaria bituminosa* (L.) C.H.Stirt.	Parnassos Mt., 2017	1.81	1.91	12.4
L07 ^2^	*Cicer arietinum* L.	Kilkis Pref., 2018	1.09	0.64	1.3
L08 ^3^	*Trifolium repens* L.	Kilkis Pref., 2018	1.28	0.85	11.9

^1^ wild *taxon*, ^2^ cultivated food, ^3^ cultivated feed.

**Table 2 antioxidants-12-00852-t002:** Polyphenolic composition, total phenolic (TPC) and tannin (TTC) contents of samples studied.

Phytochemicals	L01	L02	L03	L04	L05	L06	L07	L08
TPC ^1^	10.50 ± 0.20 *	16.20 ± 1.70 ***	64.80 ± 1.50 *	27.70 ± 0.30 *	23.50 ± 0.60 *	14.50 ± 0.20 **	10.18 ± 0.09 *	24.23 ± 0.01 ***
TTC ^2^	194.10 ± 11.90 *	tr	419.60 ± 27.40 *	tr	31.60 ± 4.10 *	2.10 ± 0.70 *	6.60 ± 2.80 *	14.90 ± 7.60 *
Procyanidin B1 ^3^	nd	nd	nd	nd	nd	nd	nd	tr
Chlorogenic Acid ^3^	0.02 ± 0.00 *	nd	1.13 ± 0.02 **	tr	tr	nd	nd	0.02 ± 0.00 *
Procyanidin B2 ^3^	nd	nd	2.23 ± 0.01 *	nd	nd	nd	nd	nd
Epicatechin ^3^	nd	nd	1.52 ± 0.06 **	nd	nd	nd	nd	nd
Epigallocatechin Gallate ^3^	nd	nd	0.02 ± 0.00 *	nd	nd	nd	nd	nd
Hesperidin ^3^	tr	nd	nd	nd	nd	nd	nd	nd
Isoquercetin ^3^	0.12 ± 0.00 **	0.16 ± 0.04 *	0.76 ± 0.00 *	0.93 ± 0.60 ***	0.17 ± 0.02 **	0.01 ± 0.00^**^	0.02 ± 0.00^**^	1.60 ± 0.20 ***
Rutin ^3^	1.8 ± 0.1 ****	0.16 ± 0.01^**^	7.79 ± 0.06 ^**^	nd	nd	nd	nd	0.05 ± 0.01 *
Quercetin ^3^	0.14 ± 0.04 ****	nd	0.11 ± 0.01 ****	0.86 ± 0.09 ****	2.30 ± 0.30 ***	nd	nd	0.30 ± 0.03 ****
Apigenin ^3^	nd	nd	0.01 ± 0.00 ***	0.01 ± 0.00 ****	0.01 ± 0.00 ***	0.01 ± 0.01 **	0.06 ± 0.02 ****	0.01 ± 0.00 ****
Kaempferol ^3^	nd	nd	nd	tr	nd	nd	0.22 ± 0.09****	tr

^1^ mg GAE (Gallic Acid Equivelant)/g extract, ^2^ mg CE (Catechin Equivelant)/g extract, ^3^ mg/g extract, nd: not determined, tr: trace. *p*-value correlating samples with corresponding standard solution: * *p* ≤ 0.005, ** *p* ≤ 0.01, *** *p* ≤ 0.03, **** *p* ≤ 0.05.

**Table 3 antioxidants-12-00852-t003:** Total carotenoid contents (TCC) and carotenoids composition of samples studied.

Carotenoids	L01	L02 *	L03 *	L04 *	L05 *	L06	L07	L08 *
TCC Hex ^1^	0.08 ± 0.00 *	1.16 ± 0.01	0.86 ± 0.01	0.76 ± 0.01	0.08 ± 0.00	5.27 ± 0.02 *	0.04 ± 0.00 *	0.10 ± 0.01
TCC DCM ^1^	3.10 ± 0.05 *	22.60 ± 0.07	5.03 ± 0.07	21.20 ± 0.30	4.77 ± 0.01	14.20 ± 0.10 *	2.54 ± 0.03 *	4.98 ± 0.06
Lutein ^2^	0.03 ± 0.00 **	0.04 ± 0.00	nd	0.05 ± 0.00	nd	0.05 ± 0.00 *	nd	nd
*β*-cryptoxanthin ^2^	nd	nd	0.02 ± 0.00	nd	nd	nd	nd	0.02 ± 0.00
*α*-carotene ^2^	0.06 ± 0.00 *	nd	0.03 ± 0.00	0.06 ± 0.00	nd	nd	nd	0.06 ± 0.00
*β*-carotene ^2^	0.09 ± 0.00 *	nd	nd	0.09 ± 0.00	nd	0.37 ± 0.00 ****	0.09 ± 0.00 ***	0.09 ± 0.00

^1^ mg *β*-CE/g extract, ^2^ mg/g extract, nd: not determined. *p*-value correlating samples with corresponding standard solution: * *p* ≤ 0.005, ** *p* ≤ 0.01, *** *p* ≤ 0.03, **** *p* ≤ 0.05.

**Table 4 antioxidants-12-00852-t004:** Outcome of DPPH (2,2-DiPhenyl-1-PicrylHydrazyl) and FRAP (Ferric Reducing Antioxidant Power) assessment results for the investigated extracts.

Sample	FRAP (mmol Fe(II)/g Extract)			DPPH (mg TE/g Extract)		
	Hex	DCM	MeOH	Hex	DCM	MeOH
L01	1.02 ± 0.03 ***	0.97 ± 0.01 **	0.14 ± 0.06 ****	<LOD	9.54 ± 0.06 **	0.23 ± 0.09 **
L02	0.71 ± 0.00 **	1.21 ± 0.01*	0.22 ± 0.00 *	1.50 ± 0.60 **	18.40 ± 0.20 ****	6.50 ± 0.40 ***
L03	0.90 ± 0.00 *	1.54 ± 0.01 ***	1.20 ± 0.40 ****	1.60 ± 0.70 *	8.60 ± 0.30 **	77.41 ± 0.07 ***
L04	0.76 ± 0.01 *	1.39 ± 0.01 *	0.51 ± 0.02 *	11.40 ± 1.10 **	11.80 ± 0.04 *	14.30 ± 0.30 **
L05	1.03 ± 0.02 ***	0.80 ± 0.00 *	0.46 ± 0.01 *	3.60 ± 0.20 **	39.03 ± 0.05 ***	18.20 ± 1.60 ***
L06	0.80 ± 0.01 ***	1.59 ± 0.00 *	0.32 ± 0.01 *	16.10 ± 0.40 *	21.04 ± 0.03 ****	5.60 ± 0.50 ***
L07	0.62 ± 0.00 **	0.76 ± 0.01 *	0.11 ± 0.00 *	<LOD	6.85 ± 0.01 *	<LOD
L08	1.03 ± 0.00 *	1.18 ± 0.00 *	0.31 ± 0.01 **	7.00 ± 1.20 **	31.92 ± 0.06 ***	6.20 ± 1.50 **

*p*-value correlating samples with corresponding standard solution: * *p* ≤ 0.005, ** *p* ≤ 0.01, *** *p* ≤ 0.03, **** *p* ≤ 0.05.

## Data Availability

Data is contained within the article and Appendix A.

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
