# Peer review of "Bioactivity of Wild and Cultivated Legumes: Phytochemical Content and Antioxidant Properties"

_antioxidants, 2023, doi:10.3390/antiox12040852_

Round 1

Reviewer 1 Report

The study of Dr Myrtsi et al., is focusing in the study of antioxidant capacity of several native Fabacae extracts. Although is an original topic since most of them have not been widely studied there are some concerns that need to be addressed:

1.       The %yield extract, even if higher using methanol,  is quite low. ¿how can it be a viable option to be added as an ingredient of animal diet, taken into account the amounts needed?

2.       No statistical analysis in table 1 and table 2. Were the experiments performed only once in the case of table 1? Please explain.

3.       Tables should include more information about the data. For example, is the mean and EEM represented or DS? In that case, what about statistical differences among them?

4.       The same comments apply for table 3 and 4

5.       Animal diet needs to ensure a high % of protein apart from antioxidant compounds. Authors need to clarify why they are proposing their value as ingredient of animal diet. ¿Are they data available regarding macronutrients?

Author Response

I am returning the revised version of our manuscript with ID number antioxidants-2262345. I appreciate the careful reading and the thoughtful comments that both referees were provided. In response to their comments, I have made the following changes (all indicated by track changes) to the manuscript that I believe address all their concerns and result in a much-improved piece of work.

In specific, the point-by-point responds to all suggestions-comments are:

 Reviewer 1 Comments:

  1. The % yield extract, even if higher using methanol, is quite low. how can it be a viable option to be added as an ingredient of animal diet, taken into account the amounts needed?

Main scope of the study is to investigate legumes usefulness as livestock forages or human diet components and not the extraction/isolation of phytochemicals. Thus, the extractions were performed for the assessment of phytochemicals contained and not for their efficient isolation. This is the reason why we did not engage in the improvement of their holistic yield for extraction. We have utilized the best extraction solvent for each class of target compound (polyphenols, carotenoids).

  1. No statistical analysis in table 1 and table 2. Were the experiments performed only once in the case of table 1? Please explain.
  2. Tables should include more information about the data. For example, is the mean and EEM represented or DS? In that case, what about statistical differences among them? 
  3. The same comments apply for table 3 and 4

Thank you for pointing out the aforementioned issues. Unfortunately, during the writing of the manuscript the relevant paragraph describing the statistical processing of our data was omitted. Into the revised version of the manuscript we have included all missing info concerning the data presented in tables 2, 3 and 4. Table 1 includes the collection data of the wild species (taxon, location) and extraction yield for each solvent performed by the best available literature solvent.

  1. Animal diet needs to ensure a high % of protein apart from antioxidant compounds. Authors need to clarify why they are proposing their value as ingredient of animal diet. Are they data available regarding macronutrients?

Legumes macronutrients’ content (protein, fat, sugars etc) have been widely studied and reported in the literature. Thus, aim of this study was the assessment of their content to bio-active secondary metabolites (polyphenols, carotenoids). We have added a comment (+literature) for this issue in the Introduction part.

Hopping that this revised version of our manuscript will prove suitable for publication, I remain,

Sincerely yours,

Serkos A. Haroutounian

Reviewer 2 Report

Manuscript presented to review with title: “Bio-functionality of wild and cultivated legumes: Phytochemical content and antioxidant properties” is very interesting and well written.

 Presented manuscript is on good scientific level and represent a very high scientific value manuscript.  

The title: Bio-functionality it is too embracing term and does not reflect the nature of the presented manuscript. In my opinion, the title should be slightly changed, for example: Characteristics of biologically active compounds of wild and cultivated legumes: Phytochemical content and antioxidant properties.

 The Abstract. Authors give a short presentation of manuscript. This section is well constructed. Please add some representative values for main results to Abstract section.

Keywords: please arrange keywords in alphabetical order.

Introduction section.

The Introduction section contains all the necessary information related to the presented topic of the article. At the end, a properly formulated research goal is included.

Materials and method section

Sub-section 2.1.2. Chemicals and Standards in this section only listed chemicals and standards should be presented. Please listed chemicals in alphabetical order for example: Hexane (HPLC/MS purity; name of producer and country and city of purchasing)…

 Results.

Table 2, page 6: In method description sub-section authors mentioned about statistical analysis. In table 2 there is no any statistical tools presented. Please add p-values as well as letters for homogeneous groups. Without statistical elaboration is not properly present results and comment it.

Table 3 and 4: similar remarks as previous to Table 2.

The discussion section presents a good comparison of the obtained results with other results available in the data basis.

Presented conclusions are corresponding with all information presented via Authors’ in manuscript text.

 General opinion:  After carefully manuscript reading, I think, that presented manuscript is a very valuable. In my opinion Manuscript should be accept with substantial (major correction), especially in statistical elaboration area,  according to my suggestions and after that publish in Antioxidants journal.

Author Response

I am returning the revised version of our manuscript with ID number antioxidants-2262345. I appreciate the careful reading and the thoughtful comments that both referees were provided. In response to their comments, I have made the following changes (all indicated by track changes) to the manuscript that I believe address all their concerns and result in a much-improved piece of work.

In specific, the point-by-point responds to all suggestions-comments are:

Reviewer 2 Comments:

 Manuscript presented to review with title: “Bio-functionality of wild and cultivated legumes: Phytochemical content and antioxidant properties” is very interesting and well written. 

 Presented manuscript is on good scientific level and represent a very high scientific value manuscript.  

The title: Bio-functionality it is too embracing term and does not reflect the nature of the presented manuscript. In my opinion, the title should be slightly changed, for example: Characteristics of biologically active compounds of wild and cultivated legumes: Phytochemical content and antioxidant properties.

We have used the specific title for our manuscript considering that main scope is the exploitation of legumes as livestock forages and human diet components with special focus on the bioactivity of contained secondary metabolites. This is the reason that we have used the term bio-functionality. In respond to reviewer’s suggestion we have modified the title to “Bioactive compounds of wild and cultivated legumes: Phytochemical content and antioxidant properties”.

The Abstract. Authors give a short presentation of manuscript. This section is well constructed. Please add some representative values for main results to Abstract section. 

Representative values have been added into Abstract section.

Keywords: please arrange keywords in alphabetical order. 

Keywords were arranged in alphabetical order

Introduction section. 

The Introduction section contains all the necessary information related to the presented topic of the article. At the end, a properly formulated research goal is included.

Materials and method section

Sub-section 2.1.2. Chemicals and Standards in this section only listed chemicals and standards should be presented. Please listed chemicals in alphabetical order for example: Hexane (HPLC/MS purity; name of producer and country and city of purchasing)…

Suggestion was incorporated.

 Results.

Table 2, page 6: In method description sub-section authors mentioned about statistical analysis. In table 2 there is no any statistical tools presented. Please add p-values as well as letters for homogeneous groups. Without statistical elaboration is not properly present results and comment it. 

Table 3 and 4: similar remarks as previous to Table 2. 

Thank you for the pointing out the issue. We have added all missing information.

The discussion section presents a good comparison of the obtained results with other results available in the data basis.

Presented conclusions are corresponding with all information presented via Authors’ in manuscript text. 

 General opinion:  After carefully manuscript reading, I think, that presented manuscript is a very valuable. In my opinion Manuscript should be accept with substantial (major correction), especially in statistical elaboration area,  according to my suggestions and after that publish in Antioxidants journal.

Hopping that this revised version of our manuscript will prove suitable for publication, I remain,

Sincerely yours,

Serkos A. Haroutounian

Round 2

Reviewer 1 Report

Thank you for the the revised version of the manuscript.

Most of the comments have been addressed. I believe the title is more appropriate and betters reflects the results of the study.

Still the data in the tables need to be improved. Statistical analysis is missing and differences are not presented. Moreover, data need to be uniformed (with respect to the decimals etc)

Author Response

Dear reviewer, 

Thank you for pointing out the important issues conquering the data presentation.  In respond to your comments we have uniformed data and presented them in Tables using the same number of decimals. In addition, we have incorporated the corresponding p-values for every sample. In supplementary material section we have provided the outcome of Bonferroni tests comparison of the determined p- values.  

We hope that this revised version of our manuscript will prove suitable for publication.

Reviewer 2 Report

Thank you for resubmitting the article after the authors' corrections. Most of my comments have been incorporated. However, there are still no statistical tools in the tables. The authors conducted a statistical analysis. However, p-value values ​​and homogeneous groups should be showed in the tables. Of course Authors declare that: “…For all calculations performed in this work, the Durbin–Watson (DW) statistical tests for the residuals and the οne-way analysis of variance (ANOVA) was used and the table indicated that the P-value was always less than 0.05…”

According to scientific rules those information should be showed in Tables. P-value was less than 0.05, but how many exactly?? For different analyzed groups or compounds individual p-value should be presented, as well as homogenous groups.

I can’t believe that for example in Table 2, row 1: TPC between objects L01 – L08 there is no statistical differences….

Other comment: Why do the last two items (L07 and L08 ) have such large decimal extensions? Previous data presented for TPC rounded to one decimal place?

Similar comment to all tables…please uniform all data to two decimal places. Not, in one case is only one decimal, but in other place is even three or four decimals.

Author Response

Dear reviewer, 

Thank you for pointing out the important issues concerning the data presentation.  In respond to your comments we have uniformed data and presented them in Tables using the same number of decimals. In addition, we have incorporated the corresponding p-values for every sample. In supplementary material section we have provided the outcome of Bonferroni tests comparison of the determined p- values.   Results reveal the correlation between the samples (low p-value of <0.05 indicates relationship between residuals and fitted values 

We hope that this revised version of our manuscript will prove suitable for publication.

Round 3

Reviewer 2 Report

The article submitted for review has already been very thoroughly corrected and checked in terms of content. All my comments on version no. 2 have been added into the text. In this situation, I consider the manuscript is ready for publication in the journal Antioxidants.